# Using safe and ethical technology to prevent and respond to sexual and interpersonal violence during adolescence and young adulthood: Identifying evidence, best practices, and pathways forward—A global scoping review protocol

**Mahboubeh Shirzad**[1]*, **Astha Ramaiya**[1], **Katie Edwards**[2], **Meng Yuan**[1], **Surabhi Bhanot**[3], **Michelle R. Kaufman**[1,3]

1 Department of Health, Behavior & Society, Bloomberg School of Public Health, Johns Hopkins University, Baltimore, Maryland, United States of America, 2 School of Social Work, University of Michigan, Ann Arbor, Michigan, United States of America, 3 Department of International Health, Bloomberg School of Public Health, Johns Hopkins University, Baltimore, Maryland, United States of America

* mshirza1@jh.edu

## Abstract

### Background

Adolescents and young adults face elevated risks of sexual and interpersonal violence (SIV) compared to adults. Technology has been used to help mitigate this issue by mapping incidents, supporting victims, and promoting behavior change to prevent re-occurrence. However, there is a risk that technology could facilitate abuse. Ensuring technology use is safe and ethical in the context of SIV is critical.

### Objective

This protocol provides a roadmap for a scoping review to synthesize global literature on how technology has been used to address SIV among adolescents and young adults. It aims to identify evidence and strategies for enhancing the safety, privacy, and effectiveness of technologies to prevent and respond to SIV. A scoping review was chosen because the evidence in this field is still emerging and fragmented. This approach allows us to map the breadth of existing research, highlight gaps, and make recommendations on the pathways forward.

### Materials and methods

A search strategy was developed to identify English-language studies published since 2008, the year of the emergence of smartphones in the United States, in databases such as PubMed, EMBASE, Web of Science, and Scopus. We will include qualitative, quantitative, and mixed-methods studies that use technology (e.g., mobile

**Data availability statement:** This study utilizes only publicly available published papers. Research results and data will be made publicly available when the study is completed and published.

**Funding:** MS Cost Center number: 1600108542 funder name: Bloomberg American Health Initiative URL: https://americanhealth.jhu.edu. The funder had no involvement in the conceptualization, design, execution, analysis, or reporting of this scoping review.

**Competing interests:** The authors have declared that no competing interests exist.

apps, websites) to address SIV among youth. This scoping review protocol was prepared following the Joanna Briggs Institute guidelines and Preferred Reporting Items for Systematic Reviews and Meta-Analyses Protocols (PRISMA-P), which will be adhered to during content analysis review process. Covidence, a web-based platform for systematic review management, will be used to manage screening and data extraction. Two independent reviewers will screen and extract data, with disagreements resolved by a third reviewer.

## Discussion

The findings of the proposed scoping review have the potential to contribute to improving technology safety, data privacy, and ethical considerations in the context of reporting and tracking SIV, alongside informing future metrics, policies, guidelines, and platform designs, contributing to the creation of more secure and effective tools for SIV reporting and tracking.

## Scoping review registration

This protocol has been registered in the Open Science Framework (OSF): https://doi.org/10.17605/OSF.IO/WUNT9

---

## Background

Gender-based violence (GBV) refers to any harmful act or threat targeting an individual or group due to their actual or perceived sex, gender identity, gender expression, sexual orientation, or sex characteristics [1]. One major category of GBV is sexual and interpersonal violence (SIV), which includes a range of behaviors such as sexual assault (e.g., non-consensual sexual contact), dating violence (e.g., controlling or abusive behavior in romantic relationships), and intimate partner violence (e.g., physical or emotional abuse by a current or former partner) [2,3]. Globally, SIV affect one in three women and girls [4], and also impacts men and boys, often in less visible ways [5,6]. For instance, in India, a study revealed that 52.4% of men reported experiencing SIV in their lifetime, with contributing factors such as lower family income, limited education, and alcohol use by perpetrators [7]. The prevalence is even higher among gender-non-binary individuals who do not place themselves in the male/female categories and may include transgender, non-conforming, third gender, and other non-cisgender individuals [8–10]. These patterns reveal the broad and complex nature of SIV affecting diverse groups. Across population segments, adolescents and youth face unique risks of SIV. In comparison to adults, adolescents are 3–4 times more likely to experience sexual violence [10–13], and approximately 10–20% adolescents become victims of sexual or physical dating violence [14]. Within the United States, 16–19 year old women are 4 times more likely to be sexually assaulted than women in other age groups [15]. Gender diverse adolescents and young adults experience violent victimization at rates four times

higher than that of cisgender young people [11,12], driven by stigma, discrimination, social isolation, and absence of protective policies [13,14]. Sexual and dating violence among youth is particularly rampant on college campuses and in peer settings, with as many as one in five women and one in ten men experiencing on-campus sexual violence at least once in their academic journeys [16–18].

The impact of technology on mitigating SIV is promising. Through the use of mobile phone applications and online platforms, crucial interventions and programs have been implemented, including mapping incidents of violence, providing information and support services to victims, conducting safety assessments, facilitating relationship support interventions, and promoting behavior change programs for perpetrators [15,19,20]. However, technology can sometimes serve as a conduit for perpetrating acts of violence. A study by the Australian Institute of Criminology indicates that approximately 75% of individuals using dating applications have encountered forms of sexual violence, such as harassment and stalking [21]. Adolescents and young adults are easy targets of technology facilitated abuse (TFA), both in partnered and non-partnered contexts. The prevalence of different forms of technology-assisted sexual assault (TASA), including image-based sexual abuse, non-consensual sexting, self-produced child sexual-abuse images, online grooming (manipulating minors for sexual purposes), revenge pornography (non-consensual sharing of intimate images), and sextortion (threats to release sexual content for more material), is as high as 15.6% among the adolescent and young adult population in the United States [17,22]. In fact, one study found that youth aged 13–17 were the age group at highest risk of being victimized by technology-assisted abuse, with most perpetrators being older than this group [17].

Safety concerns may also come up in the use of technology specifically created for addressing SIV, such as personal data misuse, breaches of personal information, and increased exposure to harm if discovered by the abusive partner [23]. Survivors may face tracking, unauthorized data sharing, or retaliation if technology is misused or insufficiently protected [24]. Ethical concerns in the use of technology are particularly relevant for young users, who may not fully understand the implications of sharing sensitive personal information or may not hold complete agency in technology use [18,25]. Developmentally inappropriate and unverified content on SIV circulating on digital platforms may further make adolescents susceptible to feelings of distress, re-traumatization, and developing incorrect perceptions of consent, victim-blaming, and healthy relationships [3,18]. Addressing these issues requires robust security, age-appropriate consent mechanisms for data collection and sharing and survivor-centered design, such as anonymized reporting and trauma-informed interfaces, to ensure protection and equitable access to digital support tools [15,26]. Hence, there is a pressing need for greater caution and deliberation in utilizing technology in safe ways to address SIV [27].

Technology-based approaches to SIV typically fall into two categories: prevention and response. Prevention efforts aim to stop violence before it occurs through strategies like public education, awareness campaigns, or behavior change tools [28]. Response strategies focus on supporting individuals after violence has occurred, including reporting systems, crisis services, and access to medical, legal, psychosocial, and recovery-oriented support [29,30]. This review considers both types of interventions to understand the full spectrum of technology's role in addressing SIV.

Ethical considerations in using technology to address SIV apply to both developers and users. Developers must ensure privacy, transparency, consent, and data protection in the design of digital tools [31,32]. Users, particularly adolescents and young adults, must be informed of their data rights and able to provide meaningful consent [31,32]. These factors are especially critical given the sensitivity of SIV and the potential for technology to both help and harm.

To optimize the safety and effectiveness of these technologies, it is essential to understand how they have supported effective prevention, response, tracking, reporting, and service delivery in various settings [33,34]. Moreover, it is important to assess whether any ethical issues have been reported during the use of such technologies, given the sensitivity of SIV. For example, some platforms have lacked sufficient anonymization or security, resulting in the unintentional disclosure of survivor identities [15]. Prioritizing robust safety and privacy measures is critical to safeguarding individuals and preventing further harm [26].

There is insufficient systematically synthesized evidence in the area of safety, privacy, and ethics for technology that seeks to address SIV, resulting in a significant knowledge gap that remains inadequately explored despite its importance [35,36]. Although a few systematic and scoping reviews have been published in recent years on the use of e-health and various technologies in addressing sexual assault, most have focused on aspects such as effectiveness, acceptability, or user experience [21,37–39]. However, no studies have systematically synthesized evidence on the safety, privacy, or ethical concerns of using technology in addressing these sensitive topics [15]. This oversight is particularly concerning given the increasing prevalence of technology-facilitated abuse, and the unintended consequences or risks associated with using technology to address SIV, posing unique challenges to survivors' safety and privacy [40] Our study fills this gap by mapping how these safety and ethical elements are integrated into technology-based interventions targeting SIV among adolescents and young adults. Given the heterogeneous, still-emerging evidence on technology-based SIV interventions, a scoping review is an ideal method [41] to map global experiences and examine how safety and privacy issues have been considered for addressing SIV. This approach allows us to systematically explore existing literature, identify gaps in knowledge, and outline areas where further research is needed [41]. By synthesizing these insights, the review will provide a comprehensive overview of global knowledge to date and offer valuable guidance for future studies to address these critical issues effectively.

### Study objectives

This scoping review protocol outlines the process for synthesizing literature, evidence, and best practices on using safe and ethical technology to address SIV. The primary objective is to examine whether safety and privacy considerations are integrated into technology for SIV prevention (e.g., awareness campaigns, educational programs) and response (e.g., survivor support services, forensic care, reporting systems). The secondary objectives are to explore how these concerns are addressed in practice, identify the types of technologies being used, assess which populations they serve, and highlight implementation challenges and opportunities to guide future research and policy.

### Materials and methods

This study will follow the guidelines for conducting scoping reviews outlined by the Joanna Briggs Institute (JBI) and in the PRISMA-P (Preferred Reporting Items for Systematic Reviews and Meta-Analyses Protocols) checklist (S1 Table) to ensure a structured approach, transparency, methodological rigor, and comprehensive reporting, increasing the reproducibility and reliability of research findings [42–44]. A scoping review was chosen because the existing literature on technology-based prevention and responses to SIV among youth is highly diverse in design, scope, and quality. This approach allows for a comprehensive overview of the available evidence and identification of key concepts, gaps, and areas for future research, making it well suited to an emerging and interdisciplinary field.

### Review questions

To explore pertinent studies and evidence, we will address the following research questions:

I. How has technology (e.g., mobile apps, websites, or software) been utilized to respond to and prevent SIV?

II. To what extent has technology been effective in facilitating incident reporting, enhancing survivor safety, and improving access to support services?

III. What ethical and safety considerations are associated with the use of technology to address SIV, including privacy, traceability, consent requirements, data storage, and data protection?

IV. How have these ethical and safety concerns been addressed in the use of technology for SIV prevention and response?

 

## Inclusion criteria

To clearly define the inclusion criteria for the proposed scoping review, we followed the Participant, Concept, Context (PCC) framework outlined in the JBI guidelines.

**Participants.** We will include published research studies focusing on adolescents and young adults, ages 10–24 of all genders, a pivotal age range when SIV rates are at their highest [39].

**Concept.** This review will include studies examining the use of technology—such as mobile applications, websites, and digital platforms—designed for SIV prevention and response. Prevention focuses on strategies, interventions, or programs aimed at stopping SIV before it occurs, while response addresses actions taken after SIV has occurred. Eligible studies may include evaluations of prevention programs, technology-assisted interventions, support and response mechanisms, policy analyses, impact assessments, qualitative research, cross-cultural studies, or education initiatives related to SIV. Given that some technology-based interventions address both the mitigation of SIV's effects and its prevention through education and awareness, this review aims to explore the full spectrum of technology's role in addressing SIV, including preventive measures, response strategies, and efforts to reduce the impact of violence.

**Context.** This review will take a global view, including studies from all countries and regions. Eligible studies must have been published in English and must cover the period from 2008 to the present, reflecting the rapid evolution of digital technologies following the launch of the first smartphones. This timeframe captures the rise of mobile and internet-based tools that have transformed how SIV is addressed. The review will consider all study designs, including qualitative, quantitative, and mixed-methods studies.

## Exclusion criteria

We will exclude studies that: (1) are not published in English, (2) do not focus on adolescents or young adults aged 10–24, (3) do not involve a technology-based intervention for SIV prevention or response, or (4) are systematic reviews, scoping reviews, narrative reviews, commentaries, short communications, proposals, theses, dissertations, posters, books, or protocols.

## Search strategy and information sources

An experienced librarian assisted in developing a comprehensive search strategy for this review (S2 Table). The strategy was designed to ensure an efficient and thorough collection of relevant literature across multiple databases. Advanced search techniques and database-specific tools were employed to capture a broad range of high-quality, credible studies.

The search, conducted in January 2025, incorporated a wide array of keywords related to SIV, such as "intimate partner violence," "gender-based violence," "interpersonal violence," "sexual abuse," "sexual assault," "sexual harassment," and "femicide." Technology-related terms included "mobile app," "mobile applications," "smartphone apps," "portable software application," "eHealth," "digital health," "telemedicine," and "technology-facilitated abuse." The full list of search terms and database-specific queries is provided in S2 Table.

To ensure the inclusivity of the search, databases like PubMed/MEDLINE, EMBASE, Scopus, and Web of Science were used. In addition, relevant gray literature sources, including UN Women, United States Agency for International Development (USAID), and Open Archives Initiative (OAIster), were included. These sources can provide valuable reports, policy briefs, and other materials that may not be published in peer-reviewed journals. This multifaceted search strategy aimed to gather diverse and high-quality resources, drawing from both published peer-reviewed literature and grey literature.

## Evidence screening and selection

After completing the database searches, all references will be imported into Covidence, a bibliographic management platform [45] to streamline the review process and enhance both the rigor and reliability of the analysis. Duplicate studies

will be removed, and two independent reviewers (MS and SB) will screen the remaining studies by title and abstract, excluding those that do not meet the inclusion criteria. Studies meeting the criteria will advance to the full-text review stage, where they will undergo a thorough examination. During this stage, the two reviewers (MS and SB) will assess the full text of each study based on the study objectives and inclusion criteria individually. Interrater reliability will be assessed and discussed if disagreements arise between the reviewers and will be resolved through discussion or with a third reviewer (MRK). Any studies that do not align with the objectives or research questions will be excluded. The review team will also pilot this process to ensure consistency in the study selection approach. We will document the reasons for excluding sources of evidence and present them in a PRISMA-ScR flow diagram. Consistent with JBI guidance for scoping reviews, we will not conduct a critical appraisal, assess methodological limitations, or evaluate the risk of bias, as the purpose of a scoping review is to provide a broad overview of existing literature rather than evaluate the quality of individual studies [43].

## Data extraction

We will extract key information and data from the selected sources of evidence based on our study aims and research questions. We developed a standardized data extraction form for this review using the JBI format to minimize bias [46] (S3 Table). The form will undergo a piloting stage, during which two reviewers (MS and SB) will test it on at least three studies to ensure accuracy, completeness, and consistency.

Extracted data will include study identification details (title, publication date, authors, location, and country), study characteristics (type and design), population characteristics (e.g., age, gender identity, sexual orientation), data collection methods (e.g., surveys, interviews), data analysis methods, the type of technology used (e.g., mobile apps, websites, software), type of violence addressed (e.g., sexual assault, dating violence), outcomes reported and type of data collected (e.g., personally identifiable information). We will also extract information on ethical and safety considerations (e.g., privacy, traceability, consent, data storage, and protection), facilitators (e.g., features or conditions that enabled effective implementation, such as usability, institutional support, or integration with services), and challenges (e.g., technical barriers, low user engagement). Summaries will emphasize limitations and recommendations for future research, policy, and practice.

## Data analysis and presentation of results

We will use basic descriptive analysis both qualitatively and quantitatively. For qualitative data, we will use reflexive content analysis to identify patterns and develop key themes across studies [47]. This method allows for flexibility in interpreting diverse data and is suitable for scoping reviews that seek to map complex topics [47,48]. For quantitative data, we will summarize frequencies and proportions and use cross-tabulations to identify patterns across study characteristics (e.g., by population group, intervention type, or technology platform). Reflexivity will be integrated throughout the process by encouraging the team to document interpretations and analytical decisions using memos, which helps to ensure transparency and minimize bias. These reflective notes will guide discussions during coding and theme refinement, supporting consistent and credible synthesis [49]. Coding and synthesis will be conducted collaboratively by two reviewers (MS and SB), who will also reflect on their positionality, including their disciplinary background and relationship to the topic. The findings will be presented as a data map, using either tables, charts, or diagrams, or in a descriptive format that aligns with the review's objectives and scope.

## Deviations from the protocol

It is common for scoping reviews to occasionally deviate from their protocols [50]. If this occurs, we will document any deviations in the final report of the scoping review, along with an explanation for why these changes were made.

**Status of the study & expected timeline**

We confirm that our study is ongoing, and data collection is not yet complete as we submit this protocol. Our search strategy is finalized, and the database search is complete. We expect to finish screening and data extraction by April 2025, with final results available in July 2025.

**Statements of ethical approval**

This study is not human subjects research, as it utilizes only publicly available published papers.

## Discussion

This scoping review protocol outlines the process for examining existing literature, evidence, and best practices on the use of safe and ethical technology in addressing SIV. It will investigate whether safety and privacy issues have been considered in technological approaches for SIV prevention and response, and how these concerns have been addressed. The findings will guide the development of tools tailored to the needs of diverse populations, offering actionable insights to inform policies, guidelines, and metrics for the ethical use of technology in SIV prevention and intervention. Recommendations will emphasize data safety, privacy, and usability, and promote technology-driven solutions that are inclusive of youth with diverse identities and experiences, and effective in both preventing SIV and supporting recovery. This includes addressing user safety and privacy concerns, enhancing access to digital tools, and ensuring that technological interventions not only provide support to survivors but also help prevent future occurrences of violence.

Additionally, the study results can help us to propose strategies to engage city, state, and national governments in improving SIV reporting protocols through education and advocacy. The review aims to support technology developers, researchers, and stakeholders in advancing digital innovations while identifying gaps in current research and implementations. By synthesizing global lessons, this review has the potential to drive more equitable, secure, and effective applications of technology in addressing SIV and supporting survivors.

## Dissemination

To ensure the findings reach relevant audiences, we plan to disseminate the results through publication in a peer-reviewed journal and presentations at academic and public health conferences. Additionally, we will share tailored policy briefs with practitioners, technology developers, and national and local government agencies involved in violence prevention. We also plan to post a summary of findings on open-access platforms, practitioner networks, and social media channels (e.g., Twitter, LinkedIn) to promote digital safety and adolescent well-being.

## Limitations

Given the comprehensive nature of these databases, the majority of relevant literature is expected to be captured, making these four databases the primary sources of data. Although relevant grey literature sources were also included, the search may still miss studies not indexed in these selected platforms, which could result in the exclusion of valuable but less visible evidence and introduce some publication or indexing bias. Moreover, only studies published in English will be considered, which may result in excluding other relevant research in other languages and introduce a potential language bias. Additionally, the expected heterogeneity in study designs, populations, technologies used, and outcomes measured may pose challenges for synthesis and may limit the comparability and generalizability of findings. Our search strategy may have missed relevant grey literature from key organizations such as ECPAT and the WeProtect Global Alliance, as the search was finalized prior to protocol submission.

## Conclusions

This scoping review protocol outlines a framework for exploring the global use of technology in addressing SIV. By analyzing the effectiveness, challenges, and ethical considerations of digital tools, it seeks to identify best practices and highlight critical gaps in existing literature. The findings aim to inform the development of evidence-based policies, guidelines, and technology designs for the U.S. context that prioritize safety, privacy, and accessibility, ultimately enhancing support systems and interventions for SIV prevention and response.

## Supporting information

**S1 Table. PRISMA-P 2015 checklist.**
(DOCX)

**S2 Table. Search strategy.**
(DOCX)

**S3 Table. Data extraction form.**
(DOCX)

## Acknowledgments

We thank Donna Hesson, MLS, at the Welch Medical Library, Johns Hopkins University, for her support in developing the search strategy for this study.

## Author contributions

**Conceptualization:** Mahboubeh Shirzad.

**Formal analysis:** Mahboubeh Shirzad.

**Funding acquisition:** Mahboubeh Shirzad.

**Investigation:** Mahboubeh Shirzad.

**Methodology:** Mahboubeh Shirzad.

**Project administration:** Mahboubeh Shirzad.

**Supervision:** Mahboubeh Shirzad, Michelle R. Kaufman.

**Visualization:** Mahboubeh Shirzad.

**Writing – original draft:** Mahboubeh Shirzad.

**Writing – review & editing:** Mahboubeh Shirzad, Astha Ramaiya, Katie Edwards, Meng Yuan, Surabhi Bhanot, Michelle R. Kaufman.

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
