## [Decision Letter · Decision Letter 0]

13 Jun 2025

Dear Dr. Shirzad,

Thank you for submitting your manuscript to PLOS ONE. After careful consideration, we feel that it has merit but does not fully meet PLOS ONE’s publication criteria as it currently stands. Therefore, we invite you to submit a revised version of the manuscript that addresses the points raised during the review process.

We look forward to receiving your revised manuscript.

Kind regards,

Morteza Arab-Zozani, Ph. D.

Academic Editor

PLOS ONE

Journal Requirements:

2. We are unable to open your Supporting Information file Archive.zip. Please kindly revise as necessary and re-upload.

Additional Editor Comments:

Dear Respectable Authors

After gathering the opinions of our reviewers, we have reached a decision regarding your article.

Our decision is: Minor Revision

Please send the response to reviewers file as soon as possible and also highlighted the changes in the main text.

Reviewers' comments:

Reviewer's Responses to Questions

**Comments to the Author**

1. Does the manuscript provide a valid rationale for the proposed study, with clearly identified and justified research questions?

Reviewer #1: Partly

Reviewer #2: Yes

Reviewer #3: Partly

Reviewer #4: Yes

2. Is the protocol technically sound and planned in a manner that will lead to a meaningful outcome and allow testing the stated hypotheses?

Reviewer #1: Partly

Reviewer #2: Yes

Reviewer #3: Yes

Reviewer #4: Yes

3. Is the methodology feasible and described in sufficient detail to allow the work to be replicable?

Reviewer #1: Yes

Reviewer #2: Yes

Reviewer #3: Yes

Reviewer #4: Yes

4. Have the authors described where all data underlying the findings will be made available when the study is complete?

Reviewer #1: No

Reviewer #2: Yes

Reviewer #3: Yes

Reviewer #4: No

5. Is the manuscript presented in an intelligible fashion and written in standard English?

Reviewer #1: Yes

Reviewer #2: Yes

Reviewer #3: Yes

Reviewer #4: Yes

You may also provide optional suggestions and comments to authors that they might find helpful in planning their study.

Reviewer #1: Dear Authors,

Thank you for the opportunity to review your protocol for the scoping review titled "Using Safe and Ethical Technology to Prevent and Respond to Sexual and Interpersonal Violence: Identifying Evidence, Best Practices, and Pathways Forward." I commend your work on addressing such a vital and timely topic.

Your protocol is well-structured, transparent, and largely adheres to PRISMA-P standards for scoping reviews. The inclusion of supplementary materials, such as the detailed search strategy and extraction form, is a significant strength and demonstrates commitment to methodological rigor.

Below, I provide detailed feedback by section, with recommendations to further enhance the clarity, completeness, and transparency of your protocol.

1. Summary (Abstract)

• The abstract provides a concise overview of the rationale, objectives, and methods.

• To further improve clarity, consider specifying the types of studies included (e.g., qualitative, quantitative, mixed-methods) directly in the summary. This will help readers quickly understand the scope and methodological breadth of your review.

2. Background and Study Objectives

• The background is comprehensive and well-referenced, clearly establishing the need for this review.

• To strengthen this section, explicitly reference any previous systematic or scoping reviews on similar topics and clarify how your review specifically addresses the existing gaps identified in previous literature. For example, detail the unique aspects of safety and ethical considerations that have not been systematically synthesized to date (aspects like data protection regulations, user consent, or risk management strategies could add depth).

• Clearly delineating primary and secondary objectives would enhance the reader’s understanding of your review’s focus areas and priorities.

3. Methods

• The methods section is generally robust, with clear descriptions of the search strategy, screening, and data extraction processes.

• Specify exclusion criteria more explicitly (e.g., non-English studies, non-youth populations, non-technology interventions). Additionally, provide a more detailed justification for the selected timeframe (2008 to present), explaining its relevance to the evolution of technology in this field.

• Clarify how grey literature or unpublished studies will be handled, as this can impact the comprehensiveness of your review.

4. Supplementary Materials

• The inclusion of a detailed search strategy and extraction form is a notable strength, supporting reproducibility and transparency.

5. Limitations

• The protocol acknowledges the limitation of including only English-language studies, which may introduce language bias. This is appropriate, but you may also wish to discuss any other potential limitations, such as the challenges in synthesizing a highly heterogeneous evidence base.

In summary, your protocol is methodologically sound and addresses a critical area of public health. Addressing the points above will further enhance its clarity, transparency, and alignment with best practices for scoping reviews. I look forward to seeing the results of your important work.

Reviewer #2: Thank you for this important forward thinking innovation to the field!

-Consider revising the title & short title to include 'youth' or 'young people' as an important descriptor.

-Review grammar in abstract Obj sentence 1.

-Consider revising the terms used. Interpersonal violence is not typically considered a sub-category of GBV as stated in line 55. Rather, interpersonal violence includes a wider range of behaviors including community (gang, etc) violence. I think the term implies a bigger scope than is envisioned here. You might mean intimate partner violence? See https://www.ncbi.nlm.nih.gov/books/NBK525208/. Consider aligning with a recent terminology pub: https://ecpat.org/wp-content/uploads/2025/04/Second-Edition-Terminology-Guidelines-final.pdf

-Please add definitions of terms such as 'online grooming', 'revenge pornography', and 'sextortion'

-Please include the full list of search terms

-Did you consider including online research on experiences of SV/GBV? It could be an important addition for the field.

-Please add a citation for the sentence: "a pivotal age range when SIV rates are at their highest before decreasing."

-Consider expanding the resources for grey/white literature to include ECPAT, WeProtect Global Alliance, and other tech-focused organizations.

-Data extraction should clearly document the type of violence addressed, type of data (if any) collected (personally identifiable information? violence outcomes? indirect outcomes? no outcomes at all?)

-What do you mean by 'facilitators' to be extracted?

-In the discussion, you talk about informing city/state governments. Since this is a global study, can 'national' governments also be added?

Reviewer #3: Please accept my review of manuscript #PONE-D-25-09607, which is being considered for publication in PLOS ONE. This manuscript outlines an evidence-based protocol with which the authors plan to conduct a scoping review. The focus of this review will be technology’s impacts on either mitigating or facilitating sexual and interpersonal violence (SIV) amongst 8-25 year olds. The protocol has many strengths, including filling a needed gap in the literature on factors influencing technology-facilitated or -mitigated SIV, responding to a rising form of SIV with both a preventative and responsive intention, as well as the rigor and depth of the proposed scoping review. In addition to these strengths, I included suggestions for improvements below, which focus on increasing clarity of wording and purpose, flow, citations, and minor suggestions on methodology. Overall, I think this is an important study and a well-devised protocol that will be ready for publication with some revisions.

Paper-wide suggestions:

Please use consistent either APA or AMA-style in-text citation formatting. AMA would include superscript numbering, while APA would include the author last names and publication date in parentheses.

General recommendation: please clarify more thoroughly, in both the introduction and method, why a scoping review is the best choice of review type for this study.

Abstract:

Please include the full version of the abbreviation PRISMA-P.

Briefly mention that Covidence is a type of software for those who are unfamiliar.

Introduction:

I have an issue with the way SIV is presented in the first paragraph. Think about and revisit.

Line 67: “age-groups” does not need a hyphen.

Line 70: “(11,12).Sexual”: insert a space.

Line 80: “Adolescents and young adults are easy target of “: change to targets.

Line 81: “both in partner and non-partner contexts”: please change to “partnered and non-partnered”

Line 82: the word “penetration” is inappropriately used in this sentence. Do you mean frequency or prevalence?

Line 85: please say “in the United States”

Line 85: “Youth aged 13-17 years were found to be at the highest risk of technology-assited abuse, with majority of the perpetrators being older than this age-group (18).”

Perhaps a more precise and grammatically correct way to word this is something like: In fact, one study found that youth aged 13-17 were the age group with the highest risk of being victimized by technology-assisted abuse, with the majority of perpetrators being older than this age group (18).

Line 94: “use(21,22)” insert a space after “use.”

Lines 94-98: Please be more specific about the “topics” that adolescents may develop “incorrect” perceptions about; also, please be more specific about what is meant by “content-based data practices” and survivor-centered “design.” I also suggest citing all of these claims.

Line 103: what is an example of a potential ethical issue that may have been reported in the past?

Lines 113-120: please be sure to cite all claims.

Method:

General recommendation: please include a description of how the “needs of diverse populations” will be methodologically approached, such as whether it will be a specific research question or piece of extracted data, as this goal is mentioned in both the introduction and discussion.

Line 135: add period.

Research question 2: This is a great question, yet it might be more scientifically fruitful to answer if it is reworded as an open question. For example, “To what extent has technology been effective in facilitating…” may give you more room to answer this question qualitatively.

The parenthetical portion of your 3rd research question is important background information about ethical considerations in SIV-related technology. I suggest integrating this information into a new paragraph in your introduction so that the reader is more aware of the possibilities regarding ethical and safety issues that are reasonable to suspect could be helpful or harmful.

Line 156: please cite.

Concept section: Similarly to my suggestion above, your description of prevention versus response strategies in this section is very helpful and would be important to address in your introduction more thoroughly to justify the focus on these types of studies.

Lines 185-186: It would read better to reword the information inside the parentheses as a separate sentence.

Lines 206-207. Please justify more fully why you will not “conduct a critical appraisal, assess methodological limitations, or evaluate the risk of bias due to the nature of study.” Every study would likely benefit from these practices, so please include more rationale for these choices. Furthermore, this statement is confusing as just after this (e.g., line 211), there is a statement regarding bias mitigation. Therefore, it would be best to either have a subsection describing such bias mitigation strategies, methodological limitations, etc., and/or more fully explain what is meant by the statement in lines 206-207.

Line 212: you write that you will conduct a pilot and refinement stage, followed by the sentence, “The research team will test it on at least three studies to confirm its accuracy and completeness.” I believe that this sentence is your description of what your piloting stage will entail, but this is slightly unclear. Please connect that this is your piloting plan, if that is correct.

Data Analysis section: please include a citation regarding your intended qualitative data analysis, and name your specific qualitative analytic approach (I believe you are describing thematic analysis). I also suggest expanding more on your plan to use reflexivity, including what the process will entail and why it is recommended and by whom.

Line 232: please cite.

Reviewer #4: I recommend for the authors to outline how they intend to share their findings from the study once it is completed.

**Do you want your identity to be public for this peer review?** For information about this choice, including consent withdrawal, please see our Privacy Policy

Reviewer #1: **Yes: ** Prof. Yordanis Enriquez Canto

Reviewer #2: **Yes: ** Ashleigh L Howard

Reviewer #3: No

Reviewer #4: No

---

## [Author Response · Author response to Decision Letter 1]

15 Jul 2025

RESPONSE MATRIX

15 July 2025

Manuscript ID: PONE-D-25-09607

Manuscript Title: Using Safe and Ethical Technology to Prevent and Respond to Sexual and Interpersonal Violence during Adolescence and Young Adulthood: Identifying Evidence, Best Practices, and Pathways Forward—A Global Scoping Review Protocol

Reviewers’ Comments (#1) Authors’ Responses (Page and line number)

1. Summary (Abstract)

• The abstract provides a concise overview of the rationale, objectives, and methods.

• To further improve clarity, consider specifying the types of studies included (e.g., qualitative, quantitative, mixed-methods) directly in the summary. This will help readers quickly understand the scope and methodological breadth of your review.

Thank you for your suggestion. We clarified the inclusion of qualitative, quantitative, and mixed-methods studies during the review. The following sentence was added: “We will include qualitative, quantitative, and mixed-methods studies.”(Abstract, lines 37–38)

2. Background and Study Objectives

• The background is comprehensive and well-referenced, clearly establishing the need for this review.

• To strengthen this section, explicitly reference any previous systematic or scoping reviews on similar topics and clarify how your review specifically addresses the existing gaps identified in previous literature. For example, detail the unique aspects of safety and ethical considerations that have not been systematically synthesized to date (aspects like data protection regulations, user consent, or risk management strategies could add depth). We cited relevant existing reviews and emphasized how our review uniquely addresses a gap in the literature. The following sentence was added “Our study fills this gap by mapping how these safety and ethical elements are integrated into technology-based interventions targeting SIV among adolescents and young adults.” (Background section, lines 135–137)

3. Clearly delineating primary and secondary objectives would enhance the reader’s understanding of your review’s focus areas and priorities. We reworded the objectives to clarify that the primary aim is to assess safety and privacy integration in SIV-related technologies, while secondary aims include exploring practical implementation, technology types, target populations, and future research needs. (Background, lines 144–152)

4. Methods

• The methods section is generally robust, with clear descriptions of the search strategy, screening, and data extraction processes.

• Specify exclusion criteria more explicitly (e.g., non-English studies, non-youth populations, non-technology interventions).

• Additionally, provide a more detailed justification for the selected timeframe (2008 to present), explaining its relevance to the evolution of technology in this field.

• Clarify how grey literature or unpublished studies will be handled, as this can impact the comprehensiveness of your review. We added explicit exclusion criteria, including non-English publications, non-youth populations, non-technology-based interventions, and non-original research. (Exclusion Criteria section, lines 199–204)

We clarified that the 2008 start date reflects the emergence of smartphones and widespread use of digital tools relevant to SIV interventions. The following sentences was added “reflecting the rapid evolution of digital technologies following the launch of the first smartphones. This timeframe captures the rise of mobile and internet-based tools that have transformed how SIV is addressed. (Context section, lines 195-198)

We clarified our approach to grey literature by explicitly listing key sources (e.g., UN Women, USAID, OAIster) and noting their value in identifying reports and materials beyond peer-reviewed journals. (Search Strategy, lines 217–220)

5. Supplementary Materials

• The inclusion of a detailed search strategy and extraction form is a notable strength, supporting reproducibility and transparency Thank you for highlighting this.

6. Limitations

• The protocol acknowledges the limitation of including only English-language studies, which may introduce language bias. This is appropriate, but you may also wish to discuss any other potential limitations, such as the challenges in synthesizing a highly heterogeneous evidence base. We expanded the Limitations section to include potential challenges related to synthesizing a highly heterogeneous evidence base. The following sentence was added “Additionally, the expected heterogeneity in study designs, populations, technologies used, and outcomes measured may pose challenges for synthesis and may limit the comparability and generalizability of findings.” (Limitations section, lines 313–315)

7.In summary, your protocol is methodologically sound and addresses a critical area of public health. Addressing the points above will further enhance its clarity, transparency, and alignment with best practices for scoping reviews. I look forward to seeing the results of your important work. Thank you for this and other helpful critiques which have strengthened our manuscript.

Reviewers’ Comments (#2) Authors’ Responses

1.Thank you for this important forward thinking innovation to the field! Consider revising the title & short title to include 'youth' or 'young people' as an important descriptor. We revised the full and short titles to include “during Adolescence and Young Adulthood.” (Title Page, Lines 2 & 4,5)

2.Review grammar in abstract Obj sentence We corrected the grammar from “addresses” to “address.” (Abstract, line 30)

3.Consider revising the terms used. Interpersonal violence is not typically considered a sub-category of GBV as stated in line 55. Rather, interpersonal violence includes a wider range of behaviors including community (gang, etc) violence. I think the term implies a bigger scope than is envisioned here. You might mean intimate partner violence? See https://www.ncbi.nlm.nih.gov/books/NBK525208/. Consider aligning with a recent terminology pub: https://ecpat.org/wp-content/uploads/2025/04/Second-Edition-Terminology-Guidelines-final.pdf

Thank you for the suggestion. We retained the original terminology "sexual and interpersonal violence (SIV)" to maintain consistency with our research framing. However, we revised the Background to clarify our conceptualization of SIV as a major category of gender-based violence (GBV), encompassing sexual assault, dating violence, and intimate partner violence. (Background, lines 57–60)

4.Please add definitions of terms such as 'online grooming', 'revenge pornography', and 'sextortion' Brief definitions were added in parentheses to clarify each term. “Online grooming (manipulating minors for sexual purposes), revenge pornography (non-consensual sharing of intimate images), and sextortion (threats to release sexual content for more material)” (Background, lines 87–89)

5.Please include the full list of search terms We added more search terms in the main text and explicitly referenced the full list in Supplementary File S2. “The search, conducted in January 2025, incorporated a wide array of keywords related to SIV, such as “intimate partner violence,” “gender-based violence,” “interpersonal violence,” “sexual abuse,” “sexual assault,” “sexual harassment,” and “femicide.” Technology-related terms included “mobile app,” “mobile applications,” “smartphone apps,” “portable software application,” “eHealth,” “digital health,” “telemedicine,” and “technology-facilitated abuse.” The full list of search terms and database-specific queries is provided in Supplementary File S2.” (Search Strategy, lines 210-215)

6.Did you consider including online research on experiences of SV/GBV? It could be an important addition for the field. Thank you for the suggestion and we agree with you. Currently, we have chosen not to include online research studies that focus solely on participants' experiences of SV/GBV outside the context of specific technological interventions. However, this is an important area of work and was noted in the discussion section of the paper.

7.Please add a citation for the sentence: "a pivotal age range when SIV rates are at their highest before decreasing." We added a citation to support this sentence in the revised manuscript. (Inclusion Criteria, Line 180)

8.Consider expanding the resources for grey/white literature to include ECPAT, WeProtect Global Alliance, and other tech-focused organizations. Thank you for the helpful suggestion. As our search was completed after submitting the study protocol, we are unable to expand the sources further. We add this as one limitation. The following sentence was added “Our search strategy may have missed relevant grey literature from key organizations such as ECPAT and the WeProtect Global Alliance, as the search was finalized prior to protocol submission.” (Limitation, Line 135-137)

9.Data extraction should clearly document the type of violence addressed, type of data (if any) collected (personally identifiable information? violence outcomes? indirect outcomes? no outcomes at all?) We revised the Data Extraction section to specify analysis methods, technology type, violence type, and data collected. (Data Extraction, lines 250–256)

10.What do you mean by 'facilitators' to be extracted? We clarified the meaning of “facilitators” as “features or conditions that enabled effective implementation.” (Data Extraction, lines 253-254)

11.In the discussion, you talk about informing city/state governments. Since this is a global study, can 'national' governments also be added? We revised the text to include "national" governments alongside city and state levels. (Discussion, line 294)

Reviewers’ Comments (#3) Authors’ Responses

1.Please use consistent either APA or AMA-style in-text citation formatting. AMA would include superscript numbering, while APA would include the author last names and publication date in parentheses. We updated all in-text citations and references to follow AMA style consistently throughout the manuscript.

2.General recommendation: please clarify more thoroughly, in both the introduction and method, why a scoping review is the best choice of review type for this study. We expanded the rationale for selecting a scoping review by explaining its suitability for mapping diverse and emerging literature, particularly in a field where concepts, technologies, and ethical considerations are still evolving. (Background, line 137; Methods, lines 158–162)

3.Abstract:

Please include the full version of the abbreviation PRISMA-P. We added the full version of the abbreviation “PRISMA-P” as “Preferred Reporting Items for Systematic Reviews and Meta-Analyses Protocols”. (Abstract, Line 40)

4.Briefly mention that Covidence is a type of software for those who are unfamiliar We added a brief clarification that Covidence is “a web-based platform for systematic review management” to assist readers unfamiliar with the tool. (Abstract, lines 41-42)

5.I have an issue with the way SIV is presented in the first paragraph. Think about and revisit. We revised the first paragraph to more clearly define SIV as a subset of GBV. The following sentence was added “One major category of GBV is sexual and interpersonal violence (SIV), which includes a range of behaviors such as sexual assault (e.g., non-consensual sexual contact), dating violence (e.g., controlling or abusive behavior in romantic relationships), and intimate partner violence (e.g., physical or emotional abuse by a current or former partner).” (Background, lines 57-60)

6.Line 67: “age-groups” does not need a hyphen. We revised the phrase “age-groups” to “age group” to correct the grammar. (Background, line 71)

7.Line 70: “(11,12).Sexual”: insert a space. We inserted a space between the citation and the word “Sexual” for proper formatting. (Background, line 74)

8.Line 80: “Adolescents and young adults are easy target of “: change to targets. We changed “target” to “targets” for grammatical accuracy. (Background, line 84)

9.Line 81: “both in partner and non-partner contexts”: please change to “partnered and non-partnered” We revised the phrase to “partnered and non-partnered” for improved clarity. (Background, line 85)

10.Line 82: the word “penetration” is inappropriately used in this sentence. Do you mean frequency or prevalence? We revised “penetration” to “prevalence” to more accurately reflect the intended meaning. (Background, line 86)

11.Line 85: please say “in the United States” We revised the sentence to specify “in the United States”. (Background, lines 90)

12.Line 85: “Youth aged 13-17 years were found to be at the highest risk of technology-assited abuse, with majority of the perpetrators being older than this age-group (18).” Perhaps a more precise and grammatically correct way to word this is something like: In fact, one study found that youth aged 13-17 were the age group with the highest risk of being victimized by technology-assisted abuse, with the majority of perpetrators being older than this age group (18). We revised the sentence for clarity and grammatical accuracy using the suggested phrasing. (Background, lines 90-92)

13.Line 94: “use(21,22)” insert a space after “use.” We inserted a space after "use" to correct the formatting. (Background, line 99)

14.Lines 94-98: Please be more specific about the “topics” that adolescents may develop “incorrect” perceptions about; also, please be more specific about what is meant by “content-based data practices” and survivor-centered “design.” I also suggest citing all of these claims. We revised the section to specify that adolescents may develop incorrect perceptions of consent, victim-blaming, and healthy relationships. We clarified “survivor-centered design” with examples such as anonymized reporting and trauma-informed interfaces and added appropriate citations. (Background, lines 101-104)

15.Line 103: What is an example of a potential ethical issue that may have been reported in the past? We added an example of ethical concerns. The following sentence was added “ For example, some platforms have lacked sufficient anonymization or security, resulting in the unintentional disclosure of survivor identities.” (Background, lines 122-124)

16.Lines 113-120: please be sure to cite all claims. We reviewed and added citations to support all claims made in this section. (Background, lines 131-143)

17.Method:

General recommendation: please include a description of how the “needs of diverse populations” will be methodologically approached, such as whether it will be a specific research question or piece of extracted data, as this goal is mentioned in both the introduction and discussion. We added “population characteristics (e.g., age, gender identity, sexual orientation) ”to the Data Extraction section (Data Extraction, lines 247-248)

18.Line 135: add period. Period added at the end of the sentence for proper punctuation. (Method, line 139)

19.Research question 2: This is a great question, yet it might be more scientifically fruitful to answer if it is reworded as an open question. For example, “To what extent has technology been effective in facilitating…” may give you more room to answer this question qualitatively. We revised Research Question 2 to be more open-ended: “To what extent has technology been effective in facilitating incident reporting, enhancing survivor safety, and improving access to support services?” (Review Questions, line 168-169)

20.The parenthetical portion of your 3rd research question is important background information about ethical considerations in SIV-related technology. I suggest integrating this information into a new paragraph in your introduction so that the reader is more aware of the possibilities regarding ethical and safety issues that are reasonable to suspect could be helpful or harmful. We moved the content regarding ethical and safety considerations from the third research question into a dedicated paragraph in the Background section to provide better context and clarity. (Background, lines 114–118)

21.Line 156: please cite. We have added an appropriate citation to support the statement. (Inclusion Criteria, line 180)

22.Concept section: Similarly to my suggestion above, your description of prevention versus response strategies in this section is

---

## [Editor Report · Decision Letter 1]

21 Jul 2025

Using Safe and Ethical Technology to Prevent and Respond to Sexual and Interpersonal Violence during Adolescence and Young Adulthood: Identifying Evidence, Best Practices, and Pathways Forward—A Global Scoping Review Protocol

PONE-D-25-09607R1

Dear Dr. Shirzad,

We’re pleased to inform you that your manuscript has been judged scientifically suitable for publication and will be formally accepted for publication once it meets all outstanding technical requirements.

Kind regards,

Morteza Arab-Zozani, Ph. D.

Academic Editor

PLOS ONE
---

## [Editor Report · Acceptance letter]

PONE-D-25-09607R1

PLOS ONE

Dear Dr. Shirzad,

I'm pleased to inform you that your manuscript has been deemed suitable for publication in PLOS ONE. Congratulations! Your manuscript is now being handed over to our production team.

Kind regards,

on behalf of

Dr. Morteza Arab-Zozani

Academic Editor

PLOS ONE